# Calibrating Accelerometer Tags with Oxygen Consumption Rate of Rainbow Trout (*Oncorhynchus mykiss*) and Their Use in Aquaculture Facility: A Case Study

**DOI:** 10.3390/ani11061496

**Published:** 2021-05-21

**Authors:** Walter Zupa, Sébastien Alfonso, Francesco Gai, Laura Gasco, Maria Teresa Spedicato, Giuseppe Lembo, Pierluigi Carbonara

**Affiliations:** 1COISPA Tecnology & Research—Experimental Station for the Study of Sea Resources, 70126 Bari, Italy; zupa@coispa.it (W.Z.); salfonso@coispa.eu (S.A.); spedicato@coispa.it (M.T.S.); lembo@coispa.it (G.L.); 2Institute of Sciences of Food Production, National Research Council, 10095 Grugliasco, Italy; francesco.gai@ispa.cnr.it; 3Department of Agricultural Forest and Food Sciences, University of Turin, 10095 Grugliasco, Italy; laura.gasco@unito.it

**Keywords:** acoustic telemetry, welfare, MO_2_, U_crit_, EMG, energetic costs

## Abstract

**Simple Summary:**

Measuring metabolic rates in free-swimming fish would provide valuable insights about the energetic costs of different life activities this is challenging to implement in the field due to the difficulty of performing such measurements. Thus, the calibration of acoustic transmitters with the oxygen consumption rate (MO_2_) could be promising to counter the limitations observed in the field. In this study, calibrations were performed in rainbow trout (*Oncorhynchus mykiss*), and a subsample of fish was implanted with such a transmitter and then followed under aquaculture conditions. The use of acoustic transmitters calibrated with MO_2_ appeared to be a promising tool to estimate energetic costs in free-swimming rainbow trout, and for welfare assessment in the aquaculture industry.

**Abstract:**

Metabolic rates are linked to the energetic costs of different activities of an animal’s life. However, measuring the metabolic rate in free-swimming fish remains challenging due to the lack of possibilities to perform these direct measurements in the field. Thus, the calibration of acoustic transmitters with the oxygen consumption rate (MO_2_) could be promising to counter these limitations. In this study, rainbow trout (*Oncorhynchus mykiss* Walbaum, 1792; n = 40) were challenged in a critical swimming test (U_crit_) to (1) obtain insights about the aerobic and anaerobic metabolism throughout electromyograms; and (2) calibrate acoustic transmitters’ signal with the MO_2_ to be later used as a proxy of energetic costs. After this calibration, the fish (n = 12) were implanted with the transmitter and were followed during ~50 days in an aquaculture facility, as a case study, to evaluate the potential of such calibration. Accelerometer data gathered from tags over a long time period were converted to estimate the MO_2_. The MO_2_ values indicated that all fish were reared under conditions that did not impact their health and welfare. In addition, a diurnal pattern with higher MO_2_ was observed for the majority of implanted trout. In conclusion, this study provides (1) biological information about the muscular activation pattern of both red and white muscle; and (2) useful tools to estimate the energetic costs in free-ranging rainbow trout. The use of acoustic transmitters calibrated with MO_2_, as a proxy of energy expenditure, could be promising for welfare assessment in the aquaculture industry.

## 1. Introduction

Fish use swimming for many activities (e.g., to escape from predators, catch prey, migrate, reproduce) and show a wide range of interspecific differences in the expression of their behaviors, which are linked to the ecology of species [1]. The oxygen consumption rate (MO_2_) during swimming activity is considered an indicator of energetic costs [2,3]. In this sense, the metabolic rates (standard metabolic rate (SMR) and maximum metabolic rate (MMR)) are important indicators of fish metabolism, as well as the aerobic scope (AS)), allowing us to estimate the scope for aerobic performances [4,5]. Briefly, the SMR represents the minimal amount of oxygen needed by a fish to support its aerobic metabolic functions under resting and post-absorptive conditions [5,6]. The MMR is considered the maximum rate of aerobic metabolism and it is extrapolated using the MO_2_ at the fish’s critical swimming speed (U_crit_) [4,7,8]. The MMR minus the closest approximation of basal metabolism (SMR) is called the aerobic scope (AS), defined as the energy available to the organism to be invested in biological processes, after fueling its standard metabolic costs [5,9]. Thus, metabolic rates are widely used in ecophysiology studies because they provide the effective aerobic cost linked to different living activities [2,5,6,10]. In particular, great interest has been oriented toward the development of indirect methods for assessing the energetic demand in free-ranging fish both in nature and aquaculture environments [11,12,13,14,15]. From this perspective, the continuous development of telemetric technology has provided a useful contribution for remotely monitoring fish in their environment via physiological tags (e.g., accelerometer tags, electromyogram tag or heart rate sensor) [13,16,17].

The electromyogram (EMG), which is defined as the bio-electrical potentials generated during the muscle contraction that are proportional, in degree and frequency, to the intensity of muscle activity [13], can measure the activity of both white (anaerobic and fast contraction) and red muscles (aerobic and slow contraction) [18]. This technology provides indirect information about their relative costs of swimming, both in terms of aerobic and anaerobic mechanisms. In addition, the EMG data have also been commonly used in the literature to study swimming behavior and behavioral patterns [13,19,20,21]. In this sense, the physiological tags are increasingly often being used in aquaculture conditions to monitor fish response to a stressor and/or rearing practices [16,22,23], showing great potential in welfare monitoring applications. Indeed, the welfare of farmed fish has recently received considerable attention (e.g., [24,25,26]). Fish welfare is overall based on the fact that fish are sentient animals capable of experiencing good or bad feelings, or emotional states [27], and so most animal welfare definitions are linked to biological functions and/or feelings [24,25]. Thus, welfare is not only related to the physiological state but also related to the overall response of the organisms in captivity [28]. In this sense, behavior and energetic balance could together represent a promising way to infer welfare in farmed fish [29,30]. By calibrating transmitters’ response with known MO_2_ values, insights about metabolic costs related to rearing practices can be remotely obtained for tagged fish. Thus, the conversion of signals recorded by the transmitter to MO_2_ values would give more relevant insights about the physiology and the energetic balance of the animal [31,32,33], valuable for welfare assessment. Moreover, the knowledge of the species-specific muscular bio-energetic models allows a more fine calibration of the physiological tag [34,35].

Thus, the aim of our study was (1) to calibrate the accelerometer tag with MO_2_ in rainbow trout (*Oncorhynchus mykiss,* Walbaum, 1792), which is one of the most important fish species of European aquaculture, and (2) to later use the data from tags as a proxy of MO_2_ in free swimming fish. Briefly, accelerometer tags are movement sensors that are continuously recording acceleration variations in time over two or three dimensions (according to the specific model of the tag) that are transmitted and stored in a receiver before further data processing. To do so, we first challenged fish in a U_crit_ test, and we estimated the metabolic variables (SMR and MMR) as the baseline of fish metabolism, and the muscular activation pattern of red and white muscles using EMG. Then, the fish were implanted with an accelerometer tag and challenged to U_crit_, and accelerometer data were calibrated as a function of MO_2_ during the test. Finally, after these calibrations, some fish were implanted with the transmitter and were followed over ~50 days in an aquaculture facility, as a case study, to evaluate the potential insights of such calibration as a tool for welfare monitoring under aquaculture conditions.

## 2. Materials and Methods

All the experiments were carried out in strict accordance with Directive 2010/63/EU. All fish manipulations were performed on fish that were completely anaesthetized (stage 4: loss of reflex activity and no reaction to strong external stimuli) to minimize pain and discomfort. The survival rate after the manipulations was 100%, and all efforts were made to minimize suffering during procedures.

### 2.1. Animals

The rainbow trout (n = 40) used in the present study for tag calibration were provided by a commercial rearing facility (Troticoltura Bassignana; Cuneo, Italy) and were left to acclimatize in our facility (COISPA Tecnologia & Ricerca, Bari, Italy) for 4 weeks before the beginning of the sensor calibration, under the same rearing conditions thereafter described. Three different groups of rainbow trout were used: the first group to measure the swimming performances and MO_2_ during the trial to estimate metabolic rates (n = 20); the second group to model both red and white muscle using EMG activity (n = 10); and finally a third group to calibrate acoustic accelerometer tags with swimming activity and the associated oxygen consumption rate, MO_2_ (n = 10). Experimental fish (mean ± standard error of body weight, BW: 427.6 ± 20.5 g; total length, TL: 32.5 ± 0.6 cm) were stocked in two 1.2 m^3^ circular fiberglass tanks under a recirculating aquaculture system (RAS) at 13 ± 1 °C and pH = 8 in freshwater a stocking density of 7 kg/m^3^ in. Fish were daily fed with a commercial feed EXTREME 6 (Veronesi, Verona, Italy; 0.8% body mass per day). During all the experiment duration, a 12:12 h light:dark cycle was maintained. Oxygen concentration was monitored continuously by automatic software Sinergia Service (Chioggia, Italy), and maintained >80% of saturation. Concentrations of NO_2_^−^ and NO_3_^−^ were monitored weekly and remained <0.15 mg/L and <100 mg/L, respectively, using commercial kit Salifert (Duiven, The Netherlands).

### 2.2. Critical Swimming Test (Ucrit) and Estimation of Metabolic Rates

U_crit_ were performed on a group of 20 rainbow trout specimens (BW: 451.80 ± 24.72 g, TL: 33.19 ± 0.65 cm) randomly selected from the batch to measure swimming performances from the 28 February 2013 to 15 April 2013. A subgroup of 10 rainbow trout, randomly selected, were used to estimate the metabolic variables (SMR, MMR and AS). For the U_crit_ tests, a 120 L Blažka-style swimming chamber was used [36]. Briefly, the chamber was made of two concentric Plexiglas tubes in which the rotation of a propeller generated a laminar water flow thanks to the presence of two plastic grids at the ends of the inner tube. The respirometer was 123 cm long with an inner diameter of 24 cm and an external one of 35 cm. This swimming chamber was built to generate a laminar water flow in the inner part of the tube [35].

Each fish was challenged individually in the respirometer. Fish were slightly anesthetized with a 30 ppm hydroalcoholic solution of clove oil to collect morphometric data (TL; BW; somatic width, SW), prior the swimming trial. Then, the fish were suddenly introduced to the swimming chamber and left for at least 60 min to acclimate to the chamber without velocity into the chamber [35]. After this acclimation time, the fish were left to react spontaneously to a slow water flow of 0.1 m/s for at least 30 min. Afterward, the water speed was increased 0.1 m/s every 10 min until the fish reached the fatigue condition, touching the rear plastic grid of the swimming chamber with the caudal fin for more than 5 s, using the same protocol as described in Zupa et al. [35].

The value of the critical swimming speed (U_crit_) was computed in accordance with the following equation [7]:(1)$$U_{crit}=U+\left(\left(\frac{t}{\Delta t}\right)*u\right)$$
where *U* represents the last speed step completed (m/s); *t* is the time elapsed in the last speed step before the fatigue condition was reached (s); ∆*t* is the duration of each complete speed step (600 s); and *u* represents the speed increment for each speed step (0.1 m/s).

For specimens with a maximum SW greater than 10% of the inner diameter of the swimming chamber, the U_crit_ value was corrected according to Smit et al. [37] according to the following formula: (2)$$U_{corrected}=U_{m}+\left(1+\left(\frac{D_{fish}}{D_{tube}}\right)\right)$$
where *U_m_* represents the speed of the water in the tube without the fish, and *D_fish_* and *D_tube_* indicate the fish maximum somatic width and the cylinder inner diameter, respectively, and both expressed in centimeters.

The relative U_crit_ (BL U_crit_) was calculated as BL U_crit_ = U_crit_/BL and expressed as body lengths per second (BL/s). The rate of oxygen uptake was assessed at each swimming speed of the U_crit_ tests with the Loligo systems’ DAQ-1 respirometer (Loligo Systems, Viborg, Denmark) using an intermittent flow respirometry approach, dividing the 10 min of the velocity step into three distinct periods. The first one consists in 5 min of “flush” (a pump injects oxygenated water into the swimming chamber; Astralpool, Brescia, Italy), continuing with 2 min of “waiting” (the pump is turned-off) before the “measuring” period, the last 3 min. The oxygen probe (a self-temperature-compensated galvanic cell) was inserted in the inner tube of the swimming chamber, continuously measuring the level of oxygen. The MO_2_ (mgO_2_/kg/h) was assessed in the closed respirometer during the last 3 min of each single speed interval, as explained above. The calculation of the oxygen consumption rate was based on the linear regression analysis of the oxygen partial pressure readings (one per second), using the software Loliresp (Loligo Systems, Viborg, Denmark) as previously described in Zupa et al. [35]. The SMR and MMR were estimated as the MO_2_ at resting (i.e., U = 0 m/s) and MO_2_ at U_crit_, respectively, using the best model describing the relationship between oxygen consumption and swimming speed (see details in the statistical analyses for model selection). In this study, the absolute AS was estimated as the difference between the MMR and SMR. Since it is generally more used in ectotherms, the absolute AS was preferred to the factorial AS, making it firstly comparable with other studies in rainbow trout [38]. In addition, in this study, the use of absolute AS allows to precisely locate the estimates of MO2 between SMR and MMR in the case study, making data gathered by the accelerometer tags more valuable.

### 2.3. Hard-Wire EMG

Hard-wire EMG analysis was performed in a subgroup of 10 fish (BW = 379.19 ± 32.89 g; TL = 31.6 ± 0.98 cm) randomly selected from the batch of rainbow trout, via the surgical implantation of electrodes (California Fine Wire Company, Grover Beach, CA, USA) in the fish muscles as described below.

Before surgical implantation, the fish were anaesthetized with a 50 ppm hydroalcoholic clove oil solution (Erbofarmosan, Bari, Italy) [39], reaching stage 4 of anesthesia (loss of reflex activity and no reaction to strong external stimuli) [40] in about 3 min. The same clove oil solution was used to irrigate gills continuously during the surgery. The wires were implanted using hypodermic needles 23G (NIPRO, Zaventem, Belgium), two in the lateral red musculature and the two others in the white muscle, both in the same spatial location but approximately 1 cm below the skin, as described in Zupa et al. [35]. The wires were introduced in the needle bending the terminal part for 3–4 mm. Each couple of electrodes, positioned at 1 cm of distance along the longitudinal axis, was then sutured to the left side of the body to minimize entanglements [35,41]. In particular, a suture was placed in the insertion point to anchor the wire while other two sutures were placed on the side of the fish, creating a loop with the wire. Sutures allowed to maintain the electrodes fixed into red and white muscle, respectively, while contrasting eventual traction movement, as shown in Zupa et al. [35]. All fish recovered successfully from the surgery.

Two electrodes, thin stainless steel (grade 304), twisted, plastic-coated wires that were 0.1 mm thick (1 m long) were inserted in the red muscle and white muscle, respectively, as described above. The two bioelectric differential signals were transferred from the muscle to the electrical amplifiers GRASS P511 (Grass Technologies, West Warwick, RI, USA) where the signal was filtered and amplified. Afterward, the bioelectrical signals passed to a computer through an AD-DAQ interface (DAQCardAI-16E-4; National Instrument, USA), where the EMG data were then stored by the Labview7 software (National Instrument, Austin, TX, USA) with a sampling rate of 5000 data per second [35]. The signal was then rectified and averaged to assess the EMG amplitude for each speed step during U_crit_ for both red and white muscles. EMG data were obtained during swimming trials carried out following the same U_crit_ protocol described in the section above.

### 2.4. Calibration of Tailbeat Accelerometer with Oxygen Consumption Rate

Finally, a third subsample of 10 rainbow trout specimens was used to calibrate the accelerometer tags with the MO_2_ (BW = 508 ± 28.5 g; TL = 35.1 ± 0.6 cm) during the swimming trials conducted using the same protocol for the critical swimming tests, as described above. The oxygen consumption was also assessed as described above.

The specimens were surgically implanted with the acoustic accelerometer V9A (Vemco, Nova Scotia; length: 43 mm; weight: 6.1 g in air and 3.3 g in water) following the procedures described in Alfonso et al. [41]. Briefly the fish were completely anaesthetized with a 50 ppm hydroalcoholic clove oil solution following the same protocol as the one described for EMG implantation. The tag was introduced into the abdominal cavity through a 1.5 cm incision close to the anal orifice (Figure 1a). The incision was closed with three independent sutures (Figure 1b). The surgery lasted 5 min on average, followed by a recovery time from anesthesia of about 10 min, before being reintroduced in their rearing tank. All fish recovered successfully from the surgery, and the surgical wounds were completely healed in a few days (i.e., 4–5 days; Figure 1c). The fish were challenged in the swimming trials at least 6 days after the surgical implantation of the tags, only in case they started again to feed spontaneously.

Concerning the analysis of accelerometric data, the sampling rate of the V9A tags was fixed to 10 Hertz (10 measurements/s) [42]. The tag returned an 8-bit value that represents the root mean square (RMS) acceleration. These values can be converted in acceleration using the following equation (acceleration (m/s^2^) = 0.01955(x), where x is the adimensional value returned by tags) resulting from the contribution of two axes (vertical and lateral directions of movement), every 30 s on average (from 15 to 45 s). This algorithm was designed to provide data with a more precise measurement of tailbeat activity, excluding the forward/backward component of the movement. Once the tag was activated and implanted in the fish’s body cavity, it began to transmit the accelerometer tag ID and the coded values corresponding to the acceleration of the tailbeat. The acceleration data were stored in the memory of submergible acoustic receiver (VR2W; Vemco, Halifax, NS, Canada), which was located in the outer part of the swimming chamber. As well as for EMG, the acceleration levels, which were stored during the swimming tests, were averaged per fish and per speed step of the U_crit_ tests.

### 2.5. Case Study

For the case study, fish (n = ~65,000) were reared in one tank (870 m^3^) from an open flow through a system in an aquaculture farm, Azienda agricola Caio (Porcia, Italy). Experimental fish (BW, 362.83 ± 58.77 g and TL: 29.89 ± 1.76 cm) were reared at a stocking density of ~27 kg/m^3^ in freshwater. At the beginning of the experiment, the fish mass was 362.8 ± 58.7 g, and the experiment lasted ~50 days. Temperature was 13 ± 1 °C, the O_2_ concentration was above 80% of saturation and pH 8. Fish were fed with EXTREME 6 (Veronesi, Verona, Italy; 0.8% body mass per day) twice a day at 8:00–9:00 and 13:00–14:00.

In the aquaculture farm, 12 fish were randomly selected and surgically implanted with accelerometer tags (V9A, Vemco, Halifax, NS, Canada) on 19 December 2012 to measure swimming activity over time. The fish selected had an average standard length of 29.89 ± 1.76 cm and an average weight of 362.83 ± 58.77 g. The fish were starved for 24 h before tag implantation and were anaesthetized with a 50 ppm hydroalcoholic clove oil solution using similar surgery protocol as described above. The same clove oil solution was used to continuously irrigate gills during the surgery. Then, the fish were released into the tanks and the swimming activity was recorded for ~50 days, starting 10 days after the surgery to allow fish to recover from the surgery. The tags used in the case study were programmed to measure, with a sampling rate of 10 Hertz (10 measurements/s). The tag ID and the coded values corresponding to the swimming activity were stored in the memory of three submergible acoustic receivers (VR2W; VEMCO, Halifax, NS, Canada) located in the tank. Data were gathered from 29 December 2012 to 17 February 2013

### 2.6. Statistical Analyses

All statistical analyses were conducted with R software [43] and were carried out at a 95% level of significance. The relationship between U_crit_ and fish BW, or TL, respectively, were tested by linear regression analysis.

Different models were tested to describe the relationship between swimming speed and oxygen consumption, red and white muscle EMG data, and accelerometer tag data. Linear, exponential, and logistic models were tested. The Akaike information criterion (AIC) was used to select the best model, based on the lowest AIC value [44]. The logistic equation used to estimate models was as follows:(3)$$y=\frac{A_{sym}}{1+e^{\frac{x_{mid}-x}{scal}}}$$
where *A_sym_* represents the curve asymptote; *x_mid_* represents the X value in correspondence of the inflection point when y = *A_sym_*/2; and *scal* is a scale parameter for the X axis. Data from one fish were discarded due to an acquisition default, resulting in n = 9 for fitting MO_2_ according to water speed. Only the best model fitting the data for each variable is presented in the results section.

For the case study, data obtained from the accelerometer tag in implanted fish were converted to MO_2_ using the calibration model elaborated from U_crit_ trials. Data from one fish were discarded due to an acquisition default, resulting in 11 fish followed during the experiment. Then, data were averaged for each day for each fish to follow the dynamic action during the experimental period using linear regressions. On the other hand, data were averaged for each hour for each fish to study the day/night pattern of MO_2_. A Wilcoxson test was applied to compare day and night period for each fish, by including the mean value for each hour. The day and night period were defined according to weather.com for the latitude at which the experiment was conducted and averaged over the experimental duration.

## 3. Results

### 3.1. Critical Swimming Speed (U_crit_) and Estimation of Metabolic Rates

The mean critical swimming speed (U_crit_) reached by trout in the experiment was 1.19 ± 0.23 m/s, which corresponds to 3.57 ± 0.51 BL/s. The individual fish performances ranged from 0.71 to 1.59 m/s, and from 2.63 to 4.57 BL/s. Positive correlations were observed between U_crit_ and BW and TL (R^2^ = 0.55 and 0.66, respectively; *p* < 0.001 for both). Additionally, correlations were observed between BLU_crit_, and BW (R^2^ = 0.14; *p* < 0.05). A weak correlation between BLU_crit_ and TL was also observed (*p* = 0.054; Table 1).

The relationship between the MO_2_ and the swimming speed was tested with three different models: linear, exponential, and logistic. Both linear and logistic models showed approximately the same AIC value while an exponential model showed a greater AIC (AIC_linear_ = 796.02; AIC_logistic_ = 796.79; AIC_exponential_ = 801.07). The logistic model was, hence, selected as best model being the logistic pattern more in accordance with the oxygen diffusion limits in tissues, as reported in Zupa et al. [35] (Figure 2; Table 2). Briefly, by selecting the sigmoid model, we assume that the reduction in the MO_2_ increase close to the U_crit_ value (i.e., curve plateau) is more biologically consistent than that described by the linear model where the increase is continuous. Thus, the sigmoid model was used to estimate the following metabolic variables: SMR, MMR, and AS. The SMR, indicating the oxygen consumption rate at rest (U = 0 m/s), was estimated to be 106.7 mgO_2_/kg/h, while the MMR (the MO_2_ extrapolated at the U_crit_ speed; U_crit_ = 1.19 m/s) was 636.7 mgO_2_/kg/h. The absolute AS, expressed as the difference between the MMR and SMR was thus estimated to be 530 mgO_2_/kg/h.

### 3.2. Muscular Activation Pattern

Concerning the muscular activity, the linear model showed the lowest AIC value for the red muscle’s EMG (AIC_linear_ = 105.08; AIC_exponential_ = 106.71; AIC_logistic_ = 107.19), while the best model for the white muscle’s EMG was the exponential model (AIC_exponential_ = −160.37; AIC_linear_ = −157.31, AIC_logistic_ = 160.36; Table 2). The results indicate that, during swimming, the red muscles of rainbow trout are constantly and linearly activated and white muscles are poorly activated at low speed while they are mainly activated at higher speed (>80% of the U_crit_; Figure 3).

### 3.3. Calibration of Accelerometer Tag with Oxygen Consumption

Finally, the calibration model of the swimming activity recorded by the accelerometer tag rate as a function of swimming speed during U_crit_ test was best represented by the exponential function (Figure 4; Table 2). The logistic model was, however, the best option to estimate the oxygen consumption as a function of the swimming activity recorded by an accelerometer tag according to the AIC criteria (Figure 5; Table 2).

In free-swimming fish implanted with the same device (i.e., Vemco V9A), the oxygen consumption rate could be estimated using the following equation:(4)$$y=\frac{675}{1+e^{\frac{43.98-x}{20.78}}}$$
where *x* is the swimming activity measured by the accelerometer tag for rainbow trout at 13 °C.

### 3.4. Case Study in Aquaculture Facility

Overall, all fish implanted with the transmitter in the aquaculture facility displayed swimming activities whose average estimated MO_2_ were well below the MMR estimated during the U_crit_ trials (297.8 ± 110.7 mgO_2_/kg/h). All fish displayed a mean estimated MO_2_ relatively close to each other with 257 mgO_2_/kg/h as a lower value for fish 6 and 348.3 mgO_2_/kg/h as a higher value for fish 9 (Figure 6). Although the estimated MO_2_ was well below the MMR, it can be observed in Figure 6 that some punctual events (represented by the tails of the distributions) in which the estimated MO_2_ increased close to reach the MMR, this could be in reference to punctual stressor events over the experimental period.

Overall, the estimated MO_2_ remained stable over the days of the experimental period in over half of the fish (54%). A significant decrease in the estimated MO_2_ over time was, however, observed in four fish (36%), and an increase in estimated MO_2_ was observed in one fish (*p* < 0.05; Figure 7). A significant decrease can reveal good acclimatization to the rearing conditions, whereas an increase suggests bad acclimatization to the rearing conditions in the facility.

As shown in Figure 8, over half of the fish displayed a diurnal pattern of activity, with an increase in the estimated MO_2_ during daytime (63%; fish 1, 2, 3, 5, 8, 10 and 11; *p* < 0.05), even if some variability in the estimated MO_2_ was observed between fish. During the experimental period, three fish did not have a significant daytime pattern (fishes 4, 6 and 7; *p* > 0.05), while only one displayed a higher MO_2_ during the night period (fish 9; *p* < 0.05; Figure 9).

## 4. Discussion

Overall, in the present study, we first evaluated the swimming performance of rainbow trout, measured the muscular activation pattern of red and white muscles, and estimated their metabolic rates and their aerobic scope during the U_crit_ test. Then, we calibrated accelerometer tags with the MO_2_ to be later used under aquaculture conditions, as a tool for health/welfare monitoring. We finally showed descriptive data from implanted free-swimming fish in the aquaculture facility, as the case study of what kind of insights can be obtained by using these transmitters calibrated with MO_2_ in the field.

First, the results of the U_crit_ trials demonstrated that, in the range of size considered (27.1–37.3 cm), both the fish total length (TL) and the body weight (BW) were significantly positively correlated with the absolute U_crit_. These results first showed that larger fish attained a higher absolute U_crit_. This was consistent with the literature which contains evidence that in fish, including salmonids, the absolute U_crit_ is expected to increase with the size of the fish [45]. Additionally, a better indicator of the higher efficiency of locomotion for larger animals was provided by the U_crit_ standardized to BL (BLU_crit_), which better described how the larger amounts of muscle mass (red and white) in larger specimens allowed for reaching a higher absolute U_crit_, thus leading to better performance in these specimens than in smaller ones. Fish with greater BL needed more energy to generate the propulsion necessary for a given swimming speed [35], and the increase in muscle mass in the bigger specimens was sufficient to reach and sustain a higher swimming performance. In the literature, a wide heterogeneity exists in either the experimental protocols (i.e., size of individuals, water temperature, rearing conditions) or the units of measurement chosen to express the U_crit_ (e.g., m/s, ft/s, or BL/s), complicating the comparison with the literature (e.g., [46,47,48]). For instance, Farrell et al. measured an average absolute U_crit_ of 0.68 m/s for rainbow trout of an average mass of 532 g at 10–15 °C [46]. At 15 °C, Alsop et al. measured an average U_crit_ of 4.52 BL/s for a juvenile trout of 10–20 g [47], while Chandroo et al. [48] and Cooke et al. [21] respectively, found absolute U_crit_ values of 0.64 and 0.90 m/s at 8.5 °C for bigger fish (~1.2 kg) [21,48]. Swimming performances appeared a bit higher compared to those cited above but overall are consistent with what has been observed for this species.

The swimming performance, as well the MO_2_, are influenced not only by intrinsic characteristics of the species but also by extrinsic factors, such as temperature, water quality or stocking density [49,50,51]. The relationship between MO_2_ and the swimming speed has been previously described in the literature for rainbow trout using different mathematical functions, mainly linear [21] and power [52]. This likely depends on the theoretical assumption that the metabolic cost of swimming increases exponentially with velocity without a limit [53]. In the present work, a logistic function provided the best fit to the experimental MO_2_, assuming that, to a certain extent, at the higher swimming speeds, the MO_2_ increase is limited by the oxygen diffusion in tissues, and metabolic rates were estimated based on that model [4,54]. In the present study, we estimated the SMR and MMR to be 106.7 and 636.7 mgO_2_/kg/h, respectively, equivalent to an AS of 530 mgO_2_/kg/h in rainbow trout at 13 °C. Estimates of metabolic rates may vary depending on rearing conditions, measurement methods and/or environmental factors [10,55], making the direct comparison between studies difficult. As an example, three-fold intraspecific change in SMR and MMR values between individuals has been already observed [10,55], with higher variations observed for MMR. Nevertheless, our estimates are within the range of what has been measured for this species (e.g., [56,57,58,59]; 42–169 mgO_2_/kg/h for SMR and 468–620 mgO_2_/kg/h for MMR at a temperature range of 10–15 °C).

In addition to the metabolic rates data, the use of EMG during swimming trials allows to assess muscle activation patterns depending on the swimming speed, and further linked the results to the oxygen consumption. Concerning aerobic metabolism, the activation of red muscle followed a linear pattern in rainbow trout, similar to what Cooke et al. [21] also observed. The results of the EMG also showed that, until ~80% of the U_crit_, the muscle activity was mainly sustained by the aerobic metabolism (i.e., red muscle). Close to the critical swimming speed, the oxygen diffusion in the tissues (aerobic metabolism) reaches its own maximum velocity, thus limiting the red muscle activation. In contrast, the white muscle was recruited according to a typical exponential pattern (i.e., the white muscle was mostly involved at more elevated swimming speeds and close to the U_crit_ value) [35,41]. The white muscles are mostly linked to the anaerobic glycolysis for energy demand, and thus they are generally used only for short bursts or for rapid acceleration: prolonged activation causes the exhaustion of these muscular fibers. Our results are consistent with the literature since Weatherley et al. [60] reported a similar pattern of EMG activity for white muscle in rainbow trout during swimming trials. In Atlantic salmon (*Salmo salar*), which is a salmonid, the red muscles are also progressively activated with the increase in swimming speeds until over 80% of the U_crit_ while the white muscles fibers were recruited only at speeds >86% of the U_crit_ [61]. Overall, this suggests quite a common activation pattern of the muscles in the salmonid family. EMG data provide valuable insights for estimating energetic costs related to swimming in rainbow trout, especially related to anaerobic metabolism.

Understanding the metabolic rates and aerobic scope of free-swimming fish is of particular interest both in the wild and aquaculture, since it provides valuable insights about the energetic costs of different life activities, and in response to environmental factors. Indeed, the fine assessment of AS would permit an accurate estimate of the relative amount of energy available to be used in response to stressors and in turn the energy life cost. The estimation of the remaining amount of energy could be used to monitor the ability of fish to compensate for environmental stress during their activities. This compensatory ability could be viewed as a measure of individual physiological state and well-being [62].

For this reason, we fine calibrated the acoustic accelerometer tags with the oxygen consumption rate (MO_2_) and the level of muscular activity. Calibrating physiological tags with swimming speed using swimming chambers has been considered by several authors as a baseline for elucidating the energetic costs of behavior in free-swimming fish [13,40,63,64]. Thus, in the present study, we implanted some fish with the transmitter and followed them during ~50 days under classic aquaculture conditions in a farm. Overall, the tagging procedure is perceived as a stressor for fishes, triggering a stress response, but the fish return to homeostasis in a few days, overall displaying normal behavior and physiology [65,66]. By recovering the tagged fish at the end of the experiment, no particular deformities or injuries were observed. In addition, previous results suggest that the tag implantation does not induce chronic stress or fish growth decrease in tagged fish [42]. Therefore, the use of such a transmitter for health/welfare monitoring in aquaculture could be promising. Overall, in this experiment, we observed that the mean estimated MO_2_ is well below the MMR over the experimental duration, giving to the fish an amount of energy available that can be used to cope with potential stressors. In this sense, we observed in the activity distributions (Figure 6), high values of MO_2_ that can be due to different factors, such as activity during feeding, digestion, social interaction, or stress events [4,67,68,69]. Then, we also monitored the MO_2_ over days of experiment. For some fish (n = 4/11), the estimated MO_2_ decreased over time, while for one fish, the MO_2_ increased and we observed no change for the other over half of the fish (n = 6/11). The increase in the MO_2_ over time could be indicative of the presence of stressors in the facility, implementing a load on the energetic budget of the fish [15,23]. While at the opposite, a decrease could well represent the acclimatization of the fish to the rearing conditions, which seems to be the case here. Finally, we used the transmitter to follow the day/night oxygen consumption pattern of the fish. Over half of the trout showed a clear higher MO_2_ during the daytime compared to nighttime, corroborating previous findings based on the direct measurement of oxygen uptake for this species [59,70], or based on swimming activity recorded by sensors [22,23,71]. These previous studies overall observed a diurnal pattern (with two peaks corresponding to feeding), however, in some cases, the trout displayed a nocturnal pattern of activity or lacked a clear pattern. This could be explained for instance by environmental factors, such as low temperature, or high stocking density that stress individuals and alter its normal behavior [21,72]. Since, in our study, the temperature cannot be considered as this low for the species, as well as the stocking density as this high, it seems that these individuals may be somewhat stressed by the rearing conditions. Interestingly, three of the four fish that did not display a diurnal pattern of MO_2_ did not decrease their MO_2_ over the experiment (i.e., fishes 4, 6 and 7). Considering these data together may indicate a potential stress state for these fish within cultures practices. More work is, however, needed to better disentangle environmental conditions that are influencing day/night activity pattern and MO_2_ in rainbow trout farming.

## 5. Conclusions

In conclusion, the AS estimation based on muscle activity and oxygen consumption offers a robust and comprehensive framework. The calibration of accelerometer tags with oxygen consumption allows the monitoring of energy cost of free-swimming rainbow trout implanted with the transmitter. This continuous monitoring over time could give valuable insights into the behavior and physiology of fishes under aquaculture conditions [21,66,73]. With the available technology (e.g., accelerometer tag, EMG), using certain activity muscle levels as an indicator of the energy cost, and thus recognizing a stress-related response more accurately is now possible, including for rainbow trout [16,49,63,71]. Although this study was descriptive, it gives valuable insights into free-swimming fish metabolism, and supports recent studies on the use of transmitters, as a promising tool for health/welfare monitoring under aquaculture conditions [22,23,69,73], and could be useful for introducing more efficient strategies for aquaculture practices.

## Figures and Tables

**Figure 1 animals-11-01496-f001:**
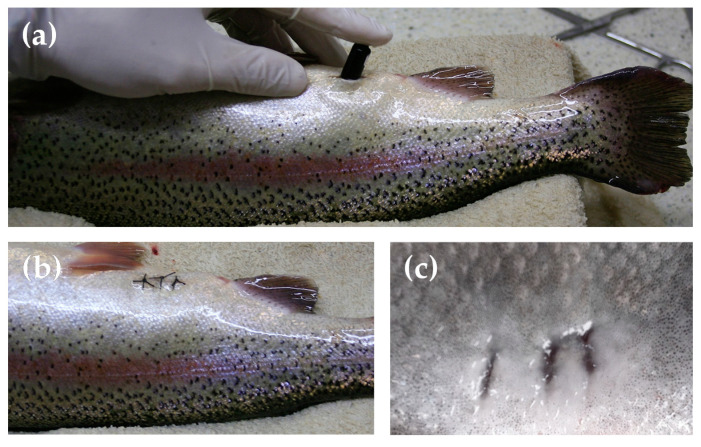
Surgical implantation of a tailbeat accelerometer tag (V9A, Vemco, Nova Scotia) in rainbow trout (*Oncorhynchus mykiss*): (**a**) insertion of the tag in the wound; (**b**) wound sutures; and (**c**) healed wound after 5 days of tag surgical implantation.

**Figure 2 animals-11-01496-f002:**
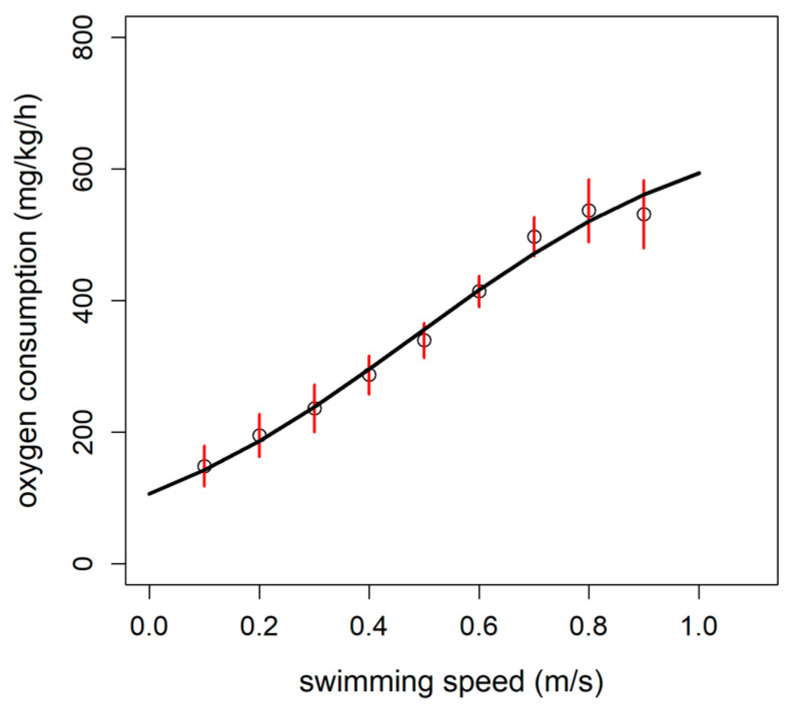
Logistic model of oxygen consumption rate (MO_2_) as a function of the swimming speed (n = 9 fish) in rainbow trout (*Oncorhynchus mykiss*). Circles with the red error bars indicate the observed mean ± standard error values, while the black line represents the prediction of the MO_2_ by the model. The estimated parameters of the logistic model were as follows: A*sym* = 668.69 ± 134.4; x*mid* = 0.48 ± 0.13; *scal* = 0.28 ± 0.07.

**Figure 3 animals-11-01496-f003:**
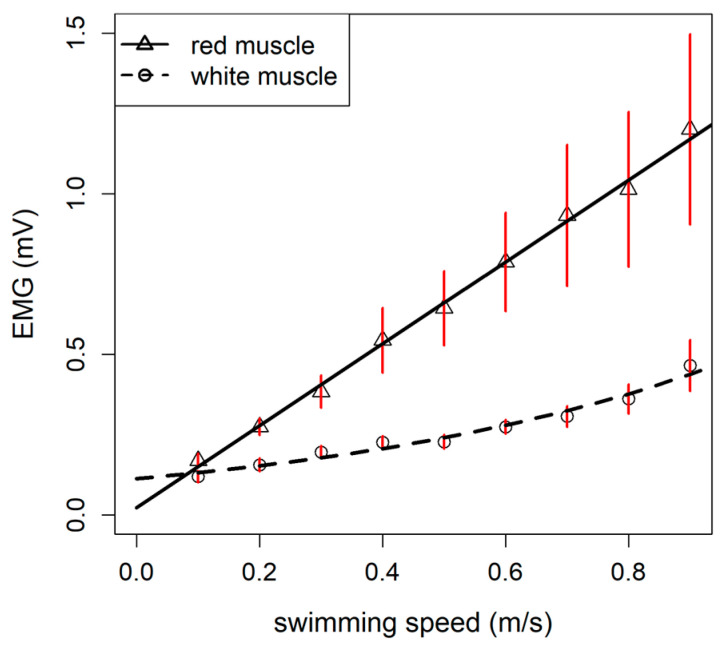
Linear model of the red muscle electromyogram (EMG) signal (continuous line; *p* < 0.05) and exponential model of the white muscle EMG signal (dashed line; *p* < 0.05) in rainbow trout (*Oncorhynchus mykiss*). Symbols (circles and triangles) with the red error bars indicate the mean ± standard error of the observed values, while the black continuous and dashed lines represent the red and white muscles’ EMG signals predicted by the models (see Table 2), respectively. The estimated parameters for the red muscle’s linear model were as follows: intercept = 0.023 ± 0.010; x coefficient = 1.275 ± 0.189 (n = 10 fish; mean ± standard error). The estimated parameters for the white muscle’s exponential model were as follows: α = 0.114 ± 0.013; β = 1.497 ± 0.166 (mean ± standard error).

**Figure 4 animals-11-01496-f004:**
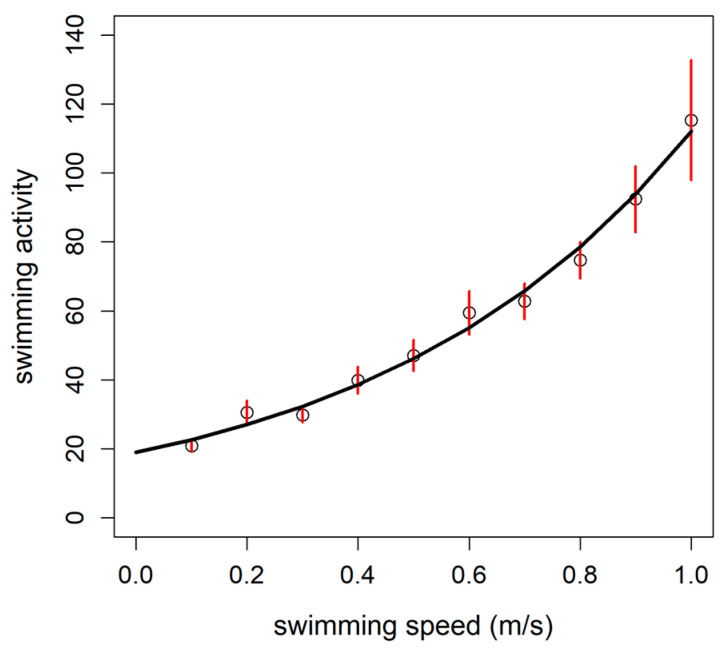
Exponential model (y = α e^βx^) of rainbow trout (*Oncorhynchus mykiss*) swimming activity as a function of the swimming speed (m/s). The estimated parameters for the exponential model were as follows: α = 18.98 ± 2.41; β = 1.77 ± 0.15 (n = 10 fish; mean ± standard error).

**Figure 5 animals-11-01496-f005:**
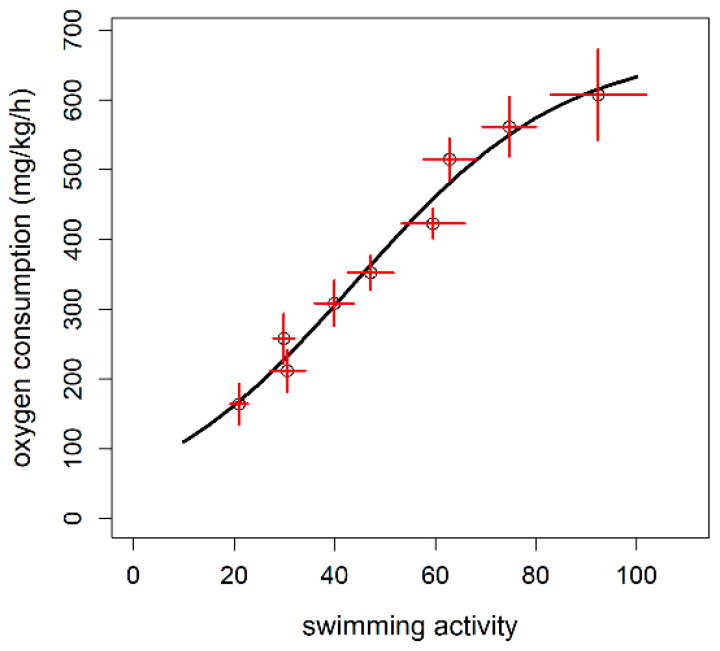
Logistic model of the oxygen consumption rate (MO_2_) as a function of the swimming activity monitored by means of the accelerometer tags in rainbow trout (*Oncorhynchus mykiss*). The estimated parameters of the logistic model were as follows: A*sym* = 675 ± 54.26; x*mid* = 43.98 ± 4.23; *scal* = 20.78 ± 3.31 (n = 10 fish; mean ± standard error).

**Figure 6 animals-11-01496-f006:**
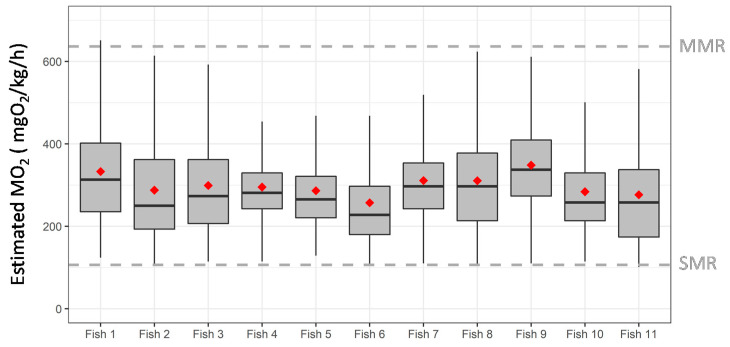
Boxplot of the estimated oxygen consumption rate (MO_2_, mgO_2_/kg/h) of implanted rainbow trout (*Oncorhynchus mykiss*) over the experimental period in aquaculture facility (case study). The central line of the boxplot indicates the median and the boxes indicate the quartiles, with the whiskers covering 95% of the values (n = 11 implanted fish). Mean value is represented for each fish with red square. The standard metabolic rate (SMR) and maximum metabolic rate (MMR), estimated during U_crit_ trials, are indicated using the dashed grey lines.

**Figure 7 animals-11-01496-f007:**
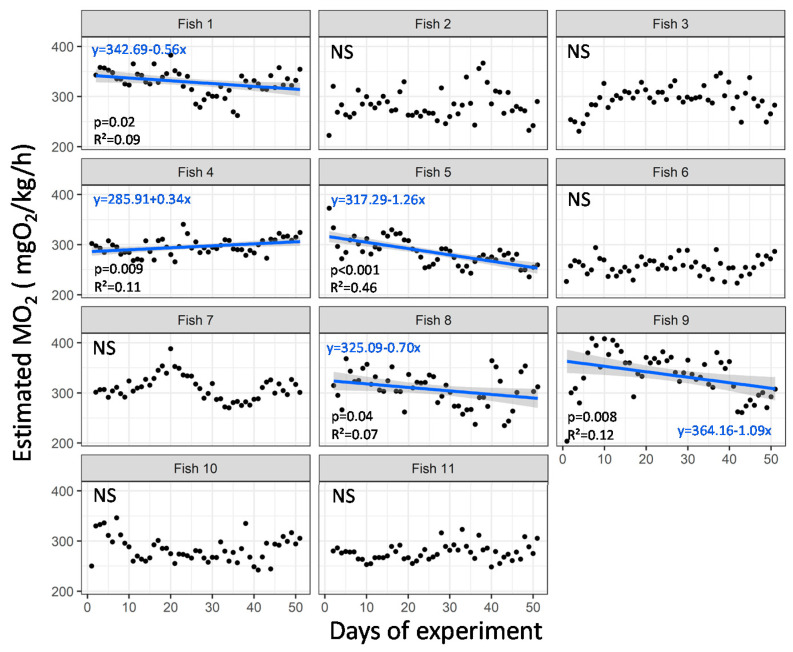
Mean of the daily estimated oxygen consumption rate (MO_2_, mgO_2_/kg/h) of implanted rainbow trout (*Oncorhynchus mykiss*) over the experimental period in aquaculture facility. The blue line represents a significant variation over time, and the grey shape represents the 95% confidence interval. R^2^ and *p* value and the coefficient of the regression are indicated for each significant linear regression (*p* < 0.05). Otherwise, NS (not significant) is indicated for each fish (n = 11 implanted fish).

**Figure 8 animals-11-01496-f008:**
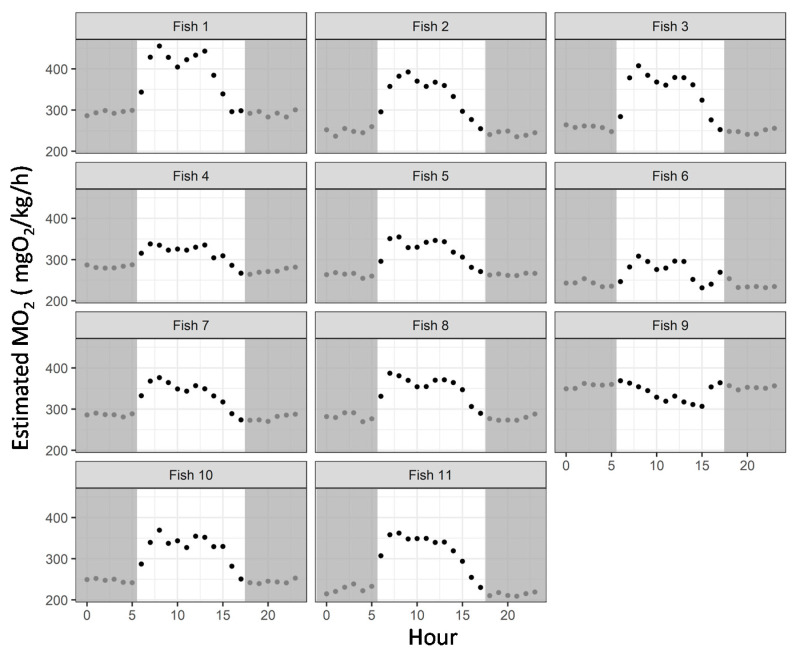
Mean of the hourly estimated oxygen consumption rate (MO_2_, mgO_2_/kg/h) of implanted rainbow trout (*Oncorhynchus mykiss*) over the day and nighttime in the aquaculture facility (n = 11 implanted fish). Dark grey shapes indicate the night period of the photoperiod.

**Figure 9 animals-11-01496-f009:**
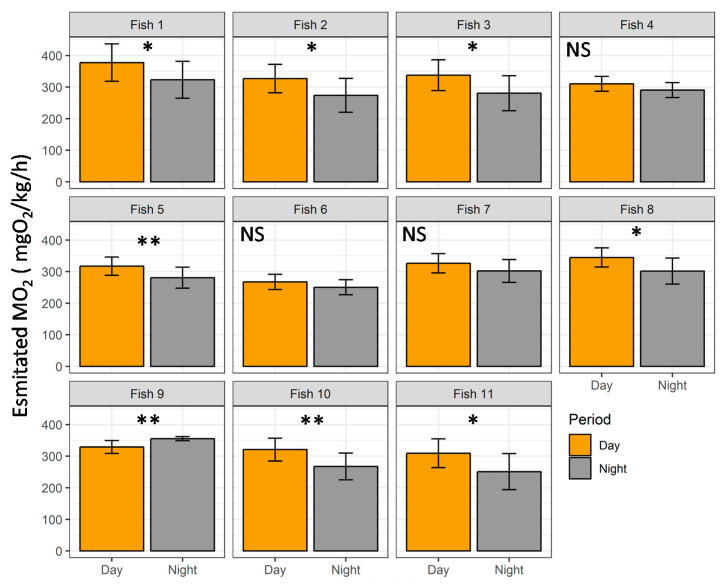
Estimated oxygen consumption rate (MO_2_, mgO_2_/kg/h) of implanted rainbow trout (*Oncorhynchus mykiss*) during the day (orange bars) and nighttime (dark grey bars) in the aquaculture facility (n = 11 implanted fish; mean ± standard error). Wilcoxon rank sum test: NS (not significant): *p* > 0.05; *: *p* < 0.05 and **: *p* < 0.01.

**Table 1 animals-11-01496-t001:** Results of the different linear regression analyses between U_crit_ and total length (TL), and body weight (BW) and between BLU_crit_ and TL, and BW in rainbow trout (*Oncorhynchus mykiss*; n = 20 fish). Estimate ± standard error of the intercept and coefficient, t value and *p* value are provided for each linear regression. R^2^ and *p* value for each significant linear regression are provided in the main text.

**U _Ucrit_**					
**Variable**	**Parameters**	**Estimate**	**Standard Error**	**t Value**	***p* Value**
TL	(intercept)	−0.79	0.40	−1.96	0.07
x		0.06	0.01	4.93	<0.001
BW	(intercept)	0.42	0.13	3.23	0.005
x		0.002	0.0003	6.13	<0.001
**BL _Ucrit_**					
**Variable**	**Parameters**	**Estimate**	**Standard Error**	**t Value**	***p* Value**
TL	(intercept)	1.06	1.22	0.86	0.40
x		0.08	0.04	2.06	0.054
BW	(intercept)	2.31	0.40	5.83	<0.001
x		0.003	0.0008	3.27	0.004

**Table 2 animals-11-01496-t002:** Summary of the parameters and statistics relative to the estimates of the different models, including the oxygen consumption rate during U_crit_, the muscular activity (red and white), and the calibration of the oxygen consumption rate with accelerometer tag in rainbow trout (*Oncorhynchus mykiss*). Estimate, standard error and associated t and *p* values were provided for each parameter of each model.

	Parameters	Estimate	St. Error	t Value	*p* Value
Ucrit–oxygen consumption rate					
	A_sym_	668.69	134.4	5.12	<0.001
	x_mid_	0.48	0.13	3.7	<0.001
	scal	0.28	0.07	3.84	<0.001
EMG–red muscle					
	intercept	0.02	0.10	0.22	0.82
	x	1.27	0.19	6.72	<0.001
EM–white muscle					
	alpha	0.11	0.01	8.87	<0.001
	beta	1.50	0.17	9.03	<0.001
Calibration–swimming activity					
	alpha	18.89	2.41	7.87	<0.001
	beta	1.78	0.16	11.43	<0.001
Calibration–oxygen consumption rate					
	A_sym_	675.45	54.26	12.45	<0.001
	x_mid_	43.98	4.23	10.39	<0.001
	scal	20.79	3.11	6.23	<0.001

## Data Availability

The data presented in this study are available on request from the corresponding author.

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
