# Peer review of "Calibrating Accelerometer Tags with Oxygen Consumption Rate of Rainbow Trout (Oncorhynchus mykiss) and Their Use in Aquaculture Facility: A Case Study"

_animals, 2021, doi:10.3390/ani11061496_

Round 1

Reviewer 1 Report

Please find below my comments for this manuscript.

Simple abstract.

L15 (‘that is challenging to be implemented in the field’) and L17 (‘to counter these limitations’), a bit vague, please reformulate

Abstract.

L23 ‘still challenging’, please develop a bit more

L25, consider mentioning the N of animals used here, since you mention it for the case study L27 (N=12)

L27, maybe rephrase, turn it around, here you used MO2 as a calibration proxy

Introduction.

L42-43, please add a reference

L51. AS definition. Maybe mention that you are here using absolute AS, as there is also the factorial AS, and depending on your species, methodology, one way to calculate the AS is more suitable.
https://link.springer.com/article/10.1007/s11160-018-9516-3

L71-72, please detail why it is necessary to use this calibration

L80, please provide a short explanation of what is an accelerometer tag

Material and methods.

Please mention the ethical permit under which the study was conducted

L89. Please provide more information regarding the facility here, Move the text from L94-101 here, then describe the laboratory experiment (L89-94), and finally the case study

L94, Table 1 and throughout the manuscript. Please change ‘total weight’ by either ‘body weight’ or just weight, and change ‘body length’ by ‘total length’. You should use for fish either total length, standard length or fork length to avoid confusion. L119, you also start to introduce Standard Length. I would recommend to only using either TL or SL throughout the manuscript, as it seems to be a bit redundant, unless you justify that the use of both method is relevant for this study (i.e some differences highlighted in table 1 between TL and SL, however, it could be due to variability in tail lengths, therefore SL may be more reliable for this species. For Salmonids, measuring fork length might be a better alternative.

L94, you are here using S.D, in other part of the manuscript, you use S.E.M, please harmonize and only use one way to represent errors. And then you only need to mention once that you provide the data as mean +/- …. In some instance, you mention what is the range (SD/SEM), in other you don’t (ex: L109).

L94 (average W: 427g), L109 (451g), L152 (379g), L176 (508g). The average weight of the fish used in the different experiments is quite different. Where the fish selected at random for those sub-experiments? Or was the last experiment conducted later in time and the fish grew? Please mention something about the selection of the fish (randomly) and/or the timeline of the experiment (i.e the experiment in the lab was conducted from XX/XX to YY/YY 20ZZ), and also when was the case study conducted. The fish metabolism also changes with the age and size of the fish. Do you take it into consideration when you calibrate the accelerometer tags, since you are using fish of 508g for this, and in the case study, you are using fish of 362 g, nearly 1/3 smaller. Please comment on that.

L99, please provide reference for this software

L101, did you also monitor ammonium values? Those are quite important as well. Regarding NO3- values, a threshold of 100 mg/L seems very high.

L106-107, perhaps provide information regarding nitrogen waste products from this facility as well, if available

Please provide complete information regarding material/software/product used (ex, L118, anesthetic used, please provide the brand + city of the manufacturer. Similar for L136-142 regarding software and pumps and L164, L168 and L169, L181, L204, L210, L220)

L146-147. Please mention how long the fish where kept at 0 m/s for the SMR  

L175-176, and elsewhere. Perhaps you should mention (also earlier), what are other alternative used while calibrating those loggers (if any) and also maybe include some sort of control group (no calibration? Different type of calibration used? to compare with your calibration method?)

L189 and figure 1. Please exchange the order of Fig b (healed wound) and c (suture); the suture should come before the healed. Also, consider removing this figure, it is not very informative and you have already 8 figures in this manuscript.

Results and Discussion

L276 and throughout the manuscript. Please harmonize how you us the fish name, sometimes you use ‘rainbow trout’, other times just ‘trout’ or ‘fish’. Similarly, please harmonize the legend of the figures/tables. Sometimes you mention ‘of rainbow trout’ and also the latin name, other times, you don’t mention the species, please choose one way and stick to it.

Fig6 (L316). I would start the Y axis on ‘0’ (it won’t be a big loss of space). Also, the upper bar for the MMR should be at the tip of the whisker for fish 1? Here it seems to be slightly below it.

L324-328, consider rephrasing. I think the term ‘majority’ could be replace by ‘over half’, since it is 6/11 fish.

Figure 7. Please provide the R values for the fish with linear regression

L335-339, Fig 8, it seems from the look of the figure that the fish 4, 7 and even 6 to certain extend also have some kind of diurnal pattern, being more active during day time. Perhaps provide the results of the tests? The ‘strange’ patterns should be discussed in the discussion. Did you see anything on the Fish 9 (already regarding it’s starting size compared with the rest of the fish?) Anything particular at the end of the experiment if you recovered the fish?

Please discuss a bit more your results, put them into perspective with the literature. For instance, L424-427 and L429-434. Please expand a bit this section. Any strange thing about those particular fish that were noted while equipping them with the loggers? Smaller fish/deformities/injuries? It could be nice to correlate your findings with other parameters (i.e plasma levels of metabolites (glucose, lactate,…). It could be hard in the aquaculture facility to recover those fish, but in the lab experiments, those are nice and easy parameters to implement, perhaps mention that it could be a good follow-up study?

Author Response

Dear Dr. Đukić,

We delighted of overall positive feedback from the two reviewers and we would like to thank them for their quick and careful check of our manuscript and valuable insights provided to improve the manuscript. We answered all comments and modified the manuscript accordingly. Changes are in “track changes” mode and lines were provided to easily track the changes. Please find bellow the letter, a point-by-point response to each comment of the reviewers.

Regarding your comments:

  • The affiliation of each authors has been carefully checked. Please find the corrected affiliations.
  • All changes were clearly highlighted in the version update on the website.
  • As already explained, all the experiments were carried in strict accordance with Directive 2010/63/EU, but the experiments were performed in 2012-2013 and this directive has only been enforced in Italy only on March, the 29th 2014. Thus, before this date, there was no need to get the number you requested to carry out the experiment, and we have no number to provide. In the revised manuscript, we added information about the dates of the experimentation. Anyhow, the protocol was approved by the Committee on the Ethics of Animal Experiments of COISPA. All fish manipulations (morphometric measurements and surgical implantations) were performed on fish that were completely anaesthetized (stage 4: loss of reflex activity and no reaction to strong external stimuli) to minimize pain and discomfort. The survival rate after the manipulations (morphometric measurements and surgical implantations) was 100%, and all efforts were made to minimize suffering.

We hope that all the responses provided are sufficient. Let us know if more details are details.

Best regards,

Pierluigi Carbonara

Also, on the behalf of co-authors

Reviewer 1

Simple abstract.

L15 (‘that is challenging to be implemented in the field’) and L17 (‘to counter these limitations’), a bit vague, please reformulate

We added precisions (ln 17 and 20).

Abstract.

L23 ‘still challenging’, please develop a bit more

We added precisions (ln 26).

L25, consider mentioning the N of animals used here, since you mention it for the case study L27 (N=12)

This information was added to the abstract (ln 30).

L27, maybe rephrase, turn it around, here you used MO2 as a calibration proxy

Some precisions were added to the sentence (ln 32).

Introduction.

L42-43, please add a reference

Following references were added to support the statement (ln 48).

Brown, J.H.; Gilooly, J.F.; Allen, A.P.; Savage, V.M.; West, G.B. Toward a metabolic theory of ecology. Ecology 2004, 85, 1771–1789, doi:10.4324/9780203114261.

Norin, T.; Clark, T.D. Measurement and relevance of maximum metabolic rate in fishes. J. Fish Biol. 2016, 88, 122–151, doi:10.1111/jfb.12796.

L51. AS definition. Maybe mention that you are here using absolute AS, as there is also the factorial AS, and depending on your species, methodology, one way to calculate the AS is more suitable.
https://link.springer.com/article/10.1007/s11160-018-9516-3

As the reviewer mentioned, we used absolute aerobic scope in this study. This is now better explained in the M&M. In addition, we also explain the rationale to use the absolute AS instead of factorial AS (see ln 179-183).

L71-72, please detail why it is necessary to use this calibration

We added a sentence to support the use of the calibration (ln 85).

L80, please provide a short explanation of what is an accelerometer tag

Brief explanation of what are accelerometer tag is now provided (ln 93).

Material and methods.

Please mention the ethical permit under which the study was conducted

All the experiments were carried out in strict accordance with Directive 2010/63/EU, but the experiments were performed in 2012-2013 and this directive has only been enforced in Italy only on March, the 29th 2014. Thus, before this date, there was no need to get the number you requested to carry out the experiment, and we have no number to provide. In the revised manuscript, we added information about the dates of the experimentation. Anyhow, the protocol was approved by the Committee on the Ethics of Animal Experiments of COISPA. All fish manipulations (morphometric measurements and surgical implantations) were performed on fish that were completely anaesthetized (stage 4: loss of reflex activity and no reaction to strong external stimuli) to minimize pain and discomfort. The survival rate after the manipulations (morphometric measurements and surgical implantations) was 100%, and all efforts were made to minimize suffering. We added ethical section in the manuscript (ln 105-109).

L89. Please provide more information regarding the facility here, Move the text from L94-101 here, then describe the laboratory experiment (L89-94), and finally the case study

We included details about the facility were fish were acclimatized and challenged for the calibration (ln 113). The text was moved according to the suggestion of the reviewer (ln 252); this is better indeed.

L94, Table 1 and throughout the manuscript. Please change ‘total weight’ by either ‘body weight’ or just weight, and change ‘body length’ by ‘total length’. You should use for fish either total length, standard length or fork length to avoid confusion. L119, you also start to introduce Standard Length. I would recommend to only using either TL or SL throughout the manuscript, as it seems to be a bit redundant, unless you justify that the use of both method is relevant for this study (i.e some differences highlighted in table 1 between TL and SL, however, it could be due to variability in tail lengths, therefore SL may be more reliable for this species. For Salmonids, measuring fork length might be a better alternative.

We corrected “total weight” to “body weight”, and “body length” to “total length” throughout the MS. Since for the case study, we only have the “total length data”, we decided to only use total length everywhere. Standard length and fork length were removed in the revised version.

L94, you are here using S.D, in other part of the manuscript, you use S.E.M, please harmonize and only use one way to represent errors. And then you only need to mention once that you provide the data as mean +/- …. In some instance, you mention what is the range (SD/SEM), in other you don’t (ex: L109).

We decided to use the standard error throughout the MS, and we mentioned only once during the first use (ln 119).

L94 (average W: 427g), L109 (451g), L152 (379g), L176 (508g). The average weight of the fish used in the different experiments is quite different. Where the fish selected at random for those sub-experiments? Or was the last experiment conducted later in time and the fish grew? Please mention something about the selection of the fish (randomly) and/or the timeline of the experiment (i.e the experiment in the lab was conducted from XX/XX to YY/YY 20ZZ), and also when was the case study conducted. The fish metabolism also changes with the age and size of the fish. Do you take it into consideration when you calibrate the accelerometer tags, since you are using fish of 508g for this, and in the case study, you are using fish of 362 g, nearly 1/3 smaller. Please comment on that.

Overall, fish were randomly selected, as already specified in the MS. This is true that there is some variation between the fish mass in fish the different experiments presented in the MS. Also, trout challenged for the calibration and trout implanted in the farm are not the same fish. Due to technical purposes, the experiment described in the case study must have been carried with smaller fish that for the calibration. We added the precise timeline when the experiments were conducted in the text (see ln 131,262 and ln 272).

As the reviewer pointed out, metabolism is progressively changing according to fish age and mass in fish. Thus, the MO2 estimated during the case study could be misestimated since size is a bit different. Following our personal experience, the metabolism is indeed changing according to fish mass but not so much for this range of mass in this species (360 vs 500 g). Further experimentations (not published works) didn’t highlight any significant variation in MO2 from 360 vs 500 g.

L99, please provide reference for this software

This reference was added to the MS (ln 125).

L101, did you also monitor ammonium values? Those are quite important as well. Regarding NO3- values, a threshold of 100 mg/L seems very high.

We did not monitor absolute ammonium values. Indeed, a value of 100 mg/l for NO3- is high but fortunately was not measured during the experiment. This value (100 mg/l) was the maximum threshold of the commercial kit. As you probably know, with these kits the values are more qualitative than quantitative, so we cannot precisely state another value than this one. With this sentence we just want to state that we monitored the NO3-  value and no issue related to NO3 - levels has been observed during our experiment. See the adds ln 127.

L106-107, perhaps provide information regarding nitrogen waste products from this facility as well, if available

Unfortunately, this information is not available.

Please provide complete information regarding material/software/product used (ex, L118, anesthetic used, please provide the brand + city of the manufacturer. Similar for L136-142 regarding software and pumps and L164, L168 and L169, L181, L204, L210, L220)

All the references are now provided in the MS.

L146-147. Please mention how long the fish where kept at 0 m/s for the SMR  

This information was already present in the MS (see ln 144).

L175-176, and elsewhere. Perhaps you should mention (also earlier), what are other alternative used while calibrating those loggers (if any) and also maybe include some sort of control group (no calibration? Different type of calibration used? to compare with your calibration method?)

This is indeed a very good suggestion from the reviewer. In this study, we cannot include some sort of control group, or different calibration used. But we definitely think about it for future studies. Anyway, in previous studies we verified that no statistical differences in physiological parameters (e.g. cortisol, lactate, glucose) and growth rate occurred between tagged and untagged fish (Alfonso 2020). Other references added to the manuscripts support the hypothesis that if the tag are less of 2% of body weight they not interfere with the normal fish physiological functions.

We added some sentences in discussion to support the use of calibration based on previous studies who did not observed particular effects of tagging (ln 508-514).

Alfonso, S.; Zupa, W.; Manfrin, A.; Fiocchi, E.; Dioguardi, M.; Dara, M.; Lembo, G.; Carbonara, P.; Cammarata, M. Surgical implantation of electronic tags does not induce medium ‑ term effect : insights from growth and stress physiological profile in two marine fish species. Anim. Biotelemetry 2020, 8, 21, doi:10.1186/s40317-020-00208-w.

L189 and figure 1. Please exchange the order of Fig b (healed wound) and c (suture); the suture should come before the healed. Also, consider removing this figure, it is not very informative and you have already 8 figures in this manuscript.

We reordered the different panels of the figure. We believe that this figure is important, since it allows to the reader to see exactly where the tag is implanting. In case it would be useful for editing issues we could anyway remove this figure.

Results and Discussion

L276 and throughout the manuscript. Please harmonize how you us the fish name, sometimes you use ‘rainbow trout’, other times just ‘trout’ or ‘fish’. Similarly, please harmonize the legend of the figures/tables. Sometimes you mention ‘of rainbow trout’ and also the latin name, other times, you don’t mention the species, please choose one way and stick to it.

We thank the reviewer for this comment. We harmonize by indicating the species and latin name both in the caption of the tables and figures.

Fig6 (L316). I would start the Y axis on ‘0’ (it won’t be a big loss of space). Also, the upper bar for the MMR should be at the tip of the whisker for fish 1? Here it seems to be slightly below it.

We redraw the figure, starting the Y axis at zero. For fish 1, some values found during the experiment were higher than the MMR, so it is normal that the line of the boxplot finish upper the MMR.

L324-328, consider rephrasing. I think the term ‘majority’ could be replace by ‘over half’, since it is 6/11 fish.

Thank for this suggestion. We modified accordingly.

Figure 7. Please provide the R values for the fish with linear regression

All the significant statistics are now provided on the figure (ln 7).

L335-339, Fig 8, it seems from the look of the figure that the fish 4, 7 and even 6 to certain extend also have some kind of diurnal pattern, being more active during day time. Perhaps provide the results of the tests? The ‘strange’ patterns should be discussed in the discussion. Did you see anything on the Fish 9 (already regarding it’s starting size compared with the rest of the fish?) Anything particular at the end of the experiment if you recovered the fish?

It seems indeed but no statistical difference has been found. We decided to only wrote p>0.05 when no significant to not overload the text. In Figure 9, we also plotted the mean day- and night time, and added the p-values for each panel. P-values are indicated in the figure as follow “*: p < 0.05 and **: p < 0.01”. We believe that it is sufficient and there is no need to write the exact p values, which will overload the figure. If the reviewer feels it necessary, we will add it.

Concerning the ‘strange’ patterns, it is now better discussed. Please see the add ln 531-539. By recovering fish at the end of the experiment, we did not find anything that can explain these results. A sentence was added to the discussion to state that more work is needed to understand the divergent patterns observed between individuals (ln 537).

Please discuss a bit more your results, put them into perspective with the literature. For instance, L424-427 and L429-434. Please expand a bit this section. Any strange thing about those particular fish that were noted while equipping them with the loggers? Smaller fish/deformities/injuries? It could be nice to correlate your findings with other parameters (i.e plasma levels of metabolites (glucose, lactate,…). It could be hard in the aquaculture facility to recover those fish, but in the lab experiments, those are nice and easy parameters to implement, perhaps mention that it could be a good follow-up study?

We did not observe any particular deformities or injuries of the tagged fish. As we previously reported for sea bass and sea bream (and explained in previous comment), tagged fish growth was similar to growth of untagged fish (Alfonso et al., 2020). This was a short communication based on a little sample size (n=5-12), but we also did not neither find any change in cortisol, glucose nor lactate levels between tagged and untagged fish. Also, other authors found that few days after the implantation of tag, fish come back to homeostasis and display normal physiology and behavior. This is indeed promising for the use of such sensors.

As suggested by the reviewer, we expand a bit this section of the discussion (see ln 508-514).

Alfonso, S.; Zupa, W.; Manfrin, A.; Fiocchi, E.; Dioguardi, M.; Dara, M.; Lembo, G.; Carbonara, P.; Cammarata, M. Surgical implantation of electronic tags does not induce medium ‑ term effect : insights from growth and stress physiological profile in two marine fish species. Anim. Biotelemetry 2020, 8, 21, doi:10.1186/s40317-020-00208-w.

Jepsen, N.; Davis, L.E.; Schreck, C.B.; Siddens, B. The Physiological Response of Chinook Salmon Smolts to Two Methods of Radio‐Tagging. Trans. Am. Fish. Soc. 2011, 130, 495–500.

Brijs, J.; Sandblom, E.; Rosengren, M.; Sundell, K.; Berg, C.; Axelsson, M.; Gräns, A. Prospects and pitfalls of using heart rate bio-loggers to assess the welfare of rainbow trout (Oncorhynchus mykiss) in aquaculture. Aquaculture 2019, 509, 188–197, doi:10.1016/j.aquaculture.2019.05.007.

Reviewer 2 Report

The ms submitted by Walter Zupa and 6 co-workers present interesting data for estimating oxygen consumption directly and indirectly in rainbow trout. The ms is relatively well written but there are numerous inaccuracies which must be corrected before publishing. I did not have any major concerns regarding the ms. I feel that the ms would benefit from professional English editing. Below I list my minor comments:

Title: It was difficult to understand and it is incorrect also. First, acoustic transmitter gives an impression of telemetry transmitters, better term would be accelometer transmitter. Second, you have not measured welfare as such but O2 consumption directly and indirectly. Try to make the title more precise, and to reflect the content of this study.

Simple summary: L14: the word “precious” (here and elsewhere) does not sound good to my ear, but that may be my problem as a non-native English speaker. I would rather use valuable. Also, change “provides” to “would provide”. L 16 (and throughout the ms): acoustic > accelometer. L 20: be careful with “welfare” as you did not measure it. If you want to use that term, you should describe what you exactly mean and how you make such inference from your data.

Abstract: L27: …, fish (n=12) were..

Introduction: L 72: species-specific

L78: MMR stated twice

L79: the rationale for using EMG is unclear if the purpose was to calibrate the accelometer tags

L97: Give manufacturer details for the feed

L102-107: move this paragraph under “case study”. What I wonder here that if this is a flow through system how it is possible, that the temperature is within 2 degrees for 50 days. Please explain or show a graph with the temperature curve.

L114: change fan to propeller and flux to flow (also elsewhere)

L115: I do not quite understand this: “The chamber volume was 120 L with a respirometer 123 cm long…”

L120: you are not saying anywhere if the fish were kept in the respirometer individually or in a group (cf. lines 109 and 111, which suggests that they were in groups but I doubt this).

L126: delete this sentence

L133: “Ucrit value was corrected according to Smit et al.”, please tell the reader why you did this correction and how you (not just the reference)

L136: please give manufacturer details (check all manufacturer information in the ms!)

L145: The name of the software is likely incorrect (apparently also in your previous article), I could not find this in Loligo’s web page. Please recheck.

L147: I think the resting speed should be U=0.1 m/s what you described earlier?

L158: hypodermic needles?

L160: ..1 cm below the skin? Not surface, I suppose. I do not quite understand. Are they positioned parallel to the long axis of the fish or how?

L161: give details of the suture used. Were the wires/electrodes (please be consistent in terminology) also implanted on the left side? Unclear, but seems to be so when I checked your earlier paper, but make it clear here. As such, this part is hard to understand properly.

L164: How long were the wires? I am also wondering, that if you stick the wire into red muscle which is just about perhaps 1-2 mm thick layer, how it is possible that the wire stays in place. Maybe I just do not understand because it is not explained properly.

L170: “5000 data per second” sounds very strange to me, would pulses (or similar) be better than data? And is it really 5000 per s, not per min? Sounds like a vast number to me.

L171: do you need to have “and each fish”? Isn’t that obvious?

L186: change the order of figures 1 b and c. 1c shows the sutures

L189: what does “a few days” mean? refer only to one figure (1c after changing the order)

L191: were there many individuals that did not start feeding?

L193: healed wound after X days of…

L205: use past tense. Delete the parenthesis.

L209: delete “see section 2.1 for details about rearing” and include the information here. Give also details how and when the fish were fed.

L212: “Fish were starved for 24 h…”

L230: why only the logistic model has been given? What does Um mean in the formula?

L235: accelometer tags

L251: Rather that “results” please be more specific. Give also the parameters (slope and constant) for the regressions. Does the inclusion of SL bring some essential information what is not enough by including BL only?

L254: delete “simultaneously”

L257-258: AIC-was lower for linear than for logistic model, so unclear why did you decided to use logistic model (cf. L228)?

L261: U=0.1 m/s? Why not BL/s?

L262: better to give as BL/s? (cf. L134-135 and 356)

Figure 2 (and also tables and other figures) heading: please try to write all headings so that they could stand alone without the need to go into text, i.e. state the species (not just “fish”), temperature and average length. L268: …indicate the observed mean … Could you also plot MO2 according to BL/s?

L274: AIC not given for logistic model, why not?

L277: “..while they are mainly…”

Table 2. L279: something wrong at the end of the line? Unclear what the parameters mean and why they are different in different cases. This is a “Hard to understand” –table without better explanation in the heading.

L281: no p-value given for the continuous line

Figure 4 and 5: what does “swimming activity” mean? No unit?

L300: use past tense

L310 and elsewhere: you must always say that “estimated average MO2” as you did not measure MO2 directly, it is just an estimate.

L312: …close to each other…

L318: ….of the estimated oxygen cons…

L321: is represented

Fig 8: you should give for each fish the ave daytime and nighttime MO2 and p-value in each panel. What strikes to my eye is that there appears two peaks during daytime for many individuals. I assume it is related to feeding, and therefore it would be important do give details how the fish were fed. L342: not just “daytime”. How is “dark period defined? Did you measure light intensity? How the day length change is taken into account in this figure? Or were the tanks inside a building without windows?

L345, 361, 367: performance

L348: later

L353: fish total length (BL)

L354: “These results first showed that larger fish attained higher absolute Ucrit” Discuss this finding with published literature.

L363-366: Despite differences (there are obviously always differences between the experiments) you should compare you findings more in detail those, rather than just saying that “results are consistent”.

L383: give the range what has been reported in the literature.

L413-414: delete “energetic demand of swimming based on”

L424: you did not monitor MO2

L438: how does it allow monitor energy costs? You have not converted your measurements to energy values.

L440: fishes

L443: including rainbow trout (delete in)

Reference list: With a quick look I found the following mistakes: Do not use capital letters in article names (e.g. L473 and elsewhere). Make sure all scientific names are in italics (all over). L552: journal name missing, L573: incorrect journal name abbreviation

Author Response

Reviewer 2

the ms submitted by Walter Zupa and 6 co-workers present interesting data for estimating oxygen consumption directly and indirectly in rainbow trout. The ms is relatively well written but there are numerous inaccuracies which must be corrected before publishing. I did not have any major concerns regarding the ms. I feel that the ms would benefit from professional English editing. Below I list my minor comments:

We would like to thank the reviewer for its review and the valuable suggestions provided for improving the MS. Please find below the responses to each comment.

Title: It was difficult to understand and it is incorrect also. First, acoustic transmitter gives an impression of telemetry transmitters, better term would be accelometer transmitter. Second, you have not measured welfare as such but O2 consumption directly and indirectly. Try to make the title more precise, and to reflect the content of this study.

According to the suggestion of the reviewer, we modified the title to “Calibrating Accelerometer Tags with Oxygen Consumption Rate of Rainbow Trout (Oncorhynchus mykiss) and their use in Aquaculture Facility: a case study” to make it more consistent with the study.

Simple summary: L14: the word “precious” (here and elsewhere) does not sound good to my ear, but that may be my problem as a non-native English speaker. I would rather use valuable. Also, change “provides” to “would provide”. L 16 (and throughout the ms): acoustic > accelometer. L 20: be careful with “welfare” as you did not measure it. If you want to use that term, you should describe what you exactly mean and how you make such inference from your data.

Valuable is better than “precious” in this context. We thank the reviewer for its suggestion and we modified accordingly. We also modified “provides” to “would provide” (ln 17). The reviewer is right, we now better specified in the introduction what we intend by welfare and why we are doing such inferences. Please see the adds ln 75-83.

Abstract: L27: …, fish (n=12) were..

Sentence was corrected (ln 33).

Introduction: L 72: species-specific

This was corrected (ln 88).

L78: MMR stated twice

This was corrected (ln 97).

L79: the rationale for using EMG is unclear if the purpose was to calibrate the accelometer tags

The reviewer is right is stating that the main purpose of this work was the accelerometer’s tag calibration. Anyway, we thought that was useful to provide also a description of the activation pattern of muscles in the species to completion of the calibration information provided. These further information provide a more complete idea of the energy allocation during swimming activity. 

L97: Give manufacturer details for the feed

This detail has been added to the MS (ln 123).

L102-107: move this paragraph under “case study”. What I wonder here that if this is a flow through system how it is possible, that the temperature is within 2 degrees for 50 days. Please explain or show a graph with the temperature curve.

This paragraph was moved in the case study section as suggested. Concerning the temperature, The temperature remains basically constant because it is taken from the well, so no drastic change has been observed during the experiment.

L114: change fan to propeller and flux to flow (also elsewhere)

We modified throughout the MS.

L115: I do not quite understand this: “The chamber volume was 120 L with a respirometer 123 cm long…”

More details about the measures of the swimming chamber were added to the manuscript (ln 137).

L120: you are not saying anywhere if the fish were kept in the respirometer individually or in a group (cf. lines 109 and 111, which suggests that they were in groups but I doubt this).

Indeed, it could be misunderstood. Fish have been kept in the respirometer and tested individually. We added this information ln 140.

L126: delete this sentence

This sentence was deleted (ln 149).

L133: “Ucrit value was corrected according to Smit et al.”, please tell the reader why you did this correction and how you (not just the reference)

We now added this information in the MS (ln 158).

L136: please give manufacturer details (check all manufacturer information in the ms!)

We provided the reference for all materials used in this study. Please see the adds, highlighted in the M&M section.

L145: The name of the software is likely incorrect (apparently also in your previous article), I could not find this in Loligo’s web page. Please recheck.

The name of the software used in this study is correct but it is not yet available on the website of the manufacturer.

L147: I think the resting speed should be U=0.1 m/s what you described earlier?

No, the SMR was estimated at resting, so when the value U=0m/s. This calculation was done using the linear model and extrapolated the value at U=0m/s, as it is actually explained in the MS ln 176.

L158: hypodermic needles?

Indeed, we added this information (ln 193).

L160: ..1 cm below the skin? Not surface, I suppose. I do not quite understand. Are they positioned parallel to the long axis of the fish or how?

We corrected “below the surface” to “below the skin”. We added more details about the positioning of the electrodes in the fish body (ln 198-202).

L161: give details of the suture used. Were the wires/electrodes (please be consistent in terminology) also implanted on the left side? Unclear, but seems to be so when I checked your earlier paper, but make it clear here. As such, this part is hard to understand properly.

Details about the suture procedures were added to clarify the method (ln 198-202; see next comment). Furthermore, about the terminology adopted, we would like to clarify that the two electrodes, as reported in the text, “Two electrodes, thin stainless steel (grade 304), twisted, plastic-coated wires that were 0.1 mm thick” (ln 204).

L164: How long were the wires? I am also wondering, that if you stick the wire into red muscle which is just about perhaps 1-2 mm thick layer, how it is possible that the wire stays in place. Maybe I just do not understand because it is not explained properly.

We added the information about the longer of the wires (ln 205).

The wires were sutured as it was specified ln 198. This suture allows the maintenance of the electrodes into red and white muscle respectively. This precision has been added in the MS for clarity (ln 200).

L170: “5000 data per second” sounds very strange to me, would pulses (or similar) be better than data? And is it really 5000 per s, not per min? Sounds like a vast number to me.

We confirm that the sampling rate has a frequency of 5000 samples per second.

L171: do you need to have “and each fish”? Isn’t that obvious?

This is indeed obvious. We deleted “and for each fish”. This also helps to alleviate the text (ln 213).

L186: change the order of figures 1 b and c. 1c shows the sutures

We followed the suggestion of the reviewer and changed the order of the figure 1.

L189: what does “a few days” mean? refer only to one figure (1c after changing the order)

This information was added and now referred only to the panel c (ln 231).

L191: were there many individuals that did not start feeding?

Issue could happen during the surgery, inducing stress to fish and so fish did not eat. In this case, fish were not tested for Ucrit and calibration. In this experiment, we did not observe any fish that did not start feeding again. But we preferred to specify here that it was monitored and that all tested fish were healthy and not stressed during the measurements.

L193: healed wound after X days of…

5days. This information was added (ln 236).

L205: use past tense. Delete the parenthesis.

We corrected accordingly (ln 248).

L209: delete “see section 2.1 for details about rearing” and include the information here. Give also details how and when the fish were fed.

We modified accordingly to the reviewer’s suggestion and added the information about feeding (ln 252-258).

L212: “Fish were starved for 24 h…”

We corrected the sentence (ln 264).

L230: why only the logistic model has been given? What does Um mean in the formula?

We only give the formula of the logistic model because it is the only one which is quite “complex”. At the contrary, we believe that exponential and linear models are well known, so we did not specify the formula in the M&M. However, in the results, the linear and exponential models are given with the parameters in each case (e.g. ln 365).

Um represents the x value to be implemented in the model. We corrected “Um” to “x” to make it easier to understand and generalize it for both oxygen consumption rate and calibration models. We thank the reviewer for this remark.

L235: accelometer tags

It was corrected (ln 288).

L251: Rather that “results” please be more specific. Give also the parameters (slope and constant) for the regressions. Does the inclusion of SL bring some essential information what is not enough by including BL only?

SL does not bring more information than only BL. We removed SL. We add all the information needed into the Table 1 (see changes ln 312).

L254: delete “simultaneously”

It was deleted (ln 314).

L257-258: AIC-was lower for linear than for logistic model, so unclear why did you decided to use logistic model (cf. L228)?

Being the difference very low for the estimated AIC values between the two models, as explained in the text (see ln 317) we selected the one with the more biological consistency. We added a sentence to justify why we decide to apply logistic instead of linear one (see ln 319-322).

L261: U=0.1 m/s? Why not BL/s?

No, it is U=0 m/s (see comment ln 147 for details). So no need to convert into BL/s.

L262: better to give as BL/s? (cf. L134-135 and 356)

Referring to the lines indicated by the referee, we have already expressed the BL Ucrit as BL/s (ln 162 and 437). It is not clear what the referee means.

Figure 2 (and also tables and other figures) heading: please try to write all headings so that they could stand alone without the need to go into text, i.e. state the species (not just “fish”), temperature and average length. L268: …indicate the observed mean … Could you also plot MO2 according to BL/s?

We now added the name of the species and the latin name in all headings. We don’t think that added the temperature, and average length is necessary. If the reviewer believes this information is needed in all captions, we will add it. As suggested, we corrected the text ln 268.

We could add this plot but this means that we also have to add supplementary statistics and model whose contribution to the completion of the work we believe is not very high. Since the range of fish total length is not wide (33.19 ± 0.65 cm), we believe that it is not necessary to overload the manuscript with this plot and associated statistics. If the reviewer feels it necessary, we will add it.

L274: AIC not given for logistic model, why not?

Thank to reviewer to point out this oversight. We added the information to the MS (ln 339).

L277: “..while they are mainly…”

The sentence was corrected (ln 341).

Table 2. L279: something wrong at the end of the line? Unclear what the parameters mean and why they are different in different cases. This is a “Hard to understand” –table without better explanation in the heading.

Indeed, there was a mistake and we corrected the end of the line (ln 345). In this table we want to report all statistics details regarding our models. We added some precisions in the heading. Let us know if more detail is needed.

L281: no p-value given for the continuous line

This was added. Exact p-values are given in Table 2 (ln 343).

Figure 4 and 5: what does “swimming activity” mean? No unit?

The swimming activity obtained by accelerometer has no unit. It can be transformed in acceleration (m/s2) by multiplying by 0.01955 according to the manufacturer instructions. This is now better specified in the material and methods (ln 240).

L300: use past tense

This was corrected (ln 369).

L310 and elsewhere: you must always say that “estimated average MO2” as you did not measure MO2 directly, it is just an estimate.

This is true. In the case study, MO2 values are estimates. We ensure that correction was applied everywhere when needed. Please see the changes in mode tracked change. Figures were also modified accordingly.

L312: …close to each other…

Sentence was corrected (ln 381).

L318: ….of the estimated oxygen cons…

This information was added in the caption. We also added in other captions for figures related to the case study (ln 388).

L321: is represented

Sentence was corrected (ln 391).

Fig 8: you should give for each fish the ave daytime and nighttime MO2 and p-value in each panel. What strikes to my eye is that there appears two peaks during daytime for many individuals. I assume it is related to feeding, and therefore it would be important do give details how the fish were fed.

We wanted like to plot the hourly data to better see the pattern over the day and night in Figure 8. As the reviewer suggested, we also plotted the mean day- and night time (see Figure 9), and added the p-values for each panel. P-values are indicated in the figure as follow “*: p < 0.05 and **: p < 0.01”. We believe that it is sufficient and there is not need to write the exact p values, which will overload the figure. If the reviewer feels it necessary, we will add it.

Concerning fish feeding, this information is very important and was added to the manuscript as suggested by the reviewer (ln 257). Indeed, these peaks are related to feeding.

L342: not just “daytime”. How is “dark period defined? Did you measure light intensity? How the day length change is taken into account in this figure? Or were the tanks inside a building without windows?

L342, we also added nighttime (see in Figure 8 caption). "Dark period" was corrected to "night" period which is more appropriated. The tanks were outside. We did not measure light intensity, but we based on day/night as defined by weather.com for the latitude at which the experiment was conducted (mean over the duration of experiment has been applied); this information was added to the manuscript (see ln 295-297).

L345, 361, 367: performance

This was corrected everywhere.

L348: later

This was corrected.

L353: fish total length (BL)

This was corrected.

L354: “These results first showed that larger fish attained higher absolute Ucrit” Discuss this finding with published literature.

This is now better discussed (ln 434).

L363-366: Despite differences (there are obviously always differences between the experiments) you should compare you findings more in detail those, rather than just saying that “results are consistent”

We now added more details of what has been reported in the literature and better compare with the literature (ln 447-452). However, as we wrote ln 445, it is difficult to compare since the temperature and size of fish is quite different.

L383: give the range what has been reported in the literature.

We added the range of what has been reported for this species in the revised MS (ln 470).

L413-414: delete “energetic demand of swimming based on”

This was deleted (ln 502).

L424: you did not monitor MO2

This is true. We corrected the sentence.

L438: how does it allow monitor energy costs? You have not converted your measurements to energy values.

This is true, we did not convert MO2 values into energetics. We thank the reviewer for its valuable remark and modulated our statement in the conclusion (ln 549).

L440: fishes

This was corrected (ln 549).

L443: including rainbow trout (delete in)

This was corrected.

Reference list: With a quick look I found the following mistakes: Do not use capital letters in article names (e.g. L473 and elsewhere). Make sure all scientific names are in italics (all over). L552: journal name missing, L573: incorrect journal name abbreviation

We checked the references list in the revised MS and ensure that all details are now present.